# FleXounDiT: Variable-Length Diffusion Transformer for Text-to-Audio Generation

## Abstract

In the real world, sounds inherently vary in length, spanning a broad spectrum of durations. We particularly aim to address the challenge of generating variable-length audio in text-to-audio (TTA) diffusion models. Extending audio length beyond what was covered during training, also known as *extrapolation*, is specifically challenging for audio generative models. The existing TTA diffusion model design notoriously suffers from the problem of generation with such flexibility. Therein, the design of prior TTA diffusion models do not accommodate to the change of positional information. In this work, we introduce a novel framework based on relative position embeddings, which is specifically designed to support the flexibility without fundamental changes to the current diffusion pipeline. Our proposed method allows *tuning-free audio length extrapolation*, thereby enhancing efficiency for generating audio with unseen lengths. Moreover, our approach enables a training strategy with shorter audio durations, enjoying reduced training costs while maintaining performance levels comparable to those achieved with longer durations. Empirically, we demonstrate exceptional performance against the existing state-of-the-arts on audio generation benchmarks with a significantly lower model size compared to the counterparts. In variable audio length generation, our approach consistently outperforms existing methods by a large margin. Our demonstration page is available at https://flexoundit.github.io/.

## 1 Introduction

Text-to-audio synthesis (TTA) is a generative task that aims to produce natural and accurate audio from text prompts (Liu et al., 2023a; Ghosal et al., 2023; Majumder et al., 2024; Huang et al., 2023b;a; Comunità et al., 2024). The applicability of TTA is reflected in assisting the sound design in the movie and game industries, accelerating creators' workflow (Li et al., 2024; Zhang et al., 2024a). Recently, diffusion models have shown a strong presence as an effective approach for text-to-audio tasks. Text-to-audio diffusion models, *e.g.*, Tango2 (Majumder et al., 2024), AudioLDM2 (Liu et al., 2023b), and Make-An-Audio2 (Huang et al., 2023b), have demonstrated notable performance in generating fixed-length audio, particularly sound effects.

Despite the success of TTA in fixed-length audio generation, real-world scenarios often require generating audio of varying lengths. As mentioned in the applicability to games and movies, the sound comes along with diverse forms and conditions, thus it would be difficult to limit audio generation only for a fixed duration. The length of each sound is influenced by the source and the environment in which it occurs, reflecting the complex and dynamic nature of the auditory landscape that surrounds us. This diversity in sound duration can range from brief, transient noises like a snap of a twig or a drop of water hitting a surface, to longer, continuous sounds *e.g.*, the roar of ocean waves, the hum of a city, or the sustained white noise. Considering the importance, the ability to generate audio of varying lengths with diffusion models remains underexplored.

To address the challenge of variable-length audio generation, we envision two possible scenarios:

- **Train-Test-Consistent-Length (TTCL).** In this setting, the model in train and test stages operates under consistent audio lengths. In other words, all target durations in testing are covered during training by means such as data augmentation. For example, if the the training data covers up to 30 seconds, the model is capable to generate variable-lengths in a range up to 30 seconds in testing.

- **Train-Short-Test-Long (TSTL).** This scenario presents shorter lengths in training but requires the model to generate longer audio lengths without further tuning, also known as *audio length extrapolation*. For instance, a model might be required to generate up to 30 seconds during testing despite that the maximum length *seen* during training is only 10 seconds. In this scenario, the model is encouraged to develop generalization capabilities due to the differences between the training and testing stages.

While TTCL shows potential in generating variable-length audio, this approach requires the model to be trained under various audio lengths that can be costly in terms of memory and time efficiency. Specifically, to generate a 30-second audio, the model must learn from 30-second audio samples during training, meaning the data must be *seen*. If we aim for a longer lengths, we must supplement the shorter audio lengths with augmentation. The TTCL scenario has been investigated by prior works (Huang et al., 2023a; Evans et al., 2024b) that limit audio generation in a certain range of duration. In contrast, the TSTL scenario provides a more efficient way to work on audio generation and allows testing on *unseen* audio lengths. TSTL allows shorter audio in training resulting a reduced number of memory to be processed and no augmented audio. TSTL would be a preferred option for training a TTA model with less computational resources. Later, we demonstrate in experiments that prior diffusion-based TTA models (Liu et al., 2023b; Majumder et al., 2024; Huang et al., 2023a) unable to preserve performance in the TSTL scenario.

To enable variable-length TTA generation while optimizing quality, flexibility, and training cost in diffusion models, we propose FleXible Sound Diffusion Transformer (FleXounDiT). Inspired by the idea of extending the tokens out of the *seen* lengths in Natural Language Processing (NLP) (Su et al., 2021; Peng et al., 2024; Chen et al., 2023; bloc97, 2023), FleXounDiT employs Diffusion Transformer (DiT) (Peebles & Xie, 2022) by introducing a novel transformer block which essentially combining the absolute position embedding and Rotary Position Embedding (RoPE) (Su et al., 2021) in the base model. Then, to address the problem in TSTL, we integrate RoPE and an improved version of Resonance YaRN (Wang et al., 2024) , allowing generation of flexible duration during the testing phase without additional finetuning. This consequently enables generating sounds of *unseen* durations. Even when trained with audio clips shorter than 10 seconds with limited memory consumption, our model guarantees high-quality sound of longer durations, which distinguishes our model from prior arts that requires long-duration training data.

In summary, our contributions in this paper are:

1. **Flexibility**. FleXounDiT, a TTA model, generates sound events with consistently high quality across variable durations. We showcase the "train-short-test-long" (TSTL) scenario, revealing that to apply for unseen long durations our approach is finetuning-free.

2. **Performance**. FleXounDiT not only generates variable-length audios but also surpasses the state-of-the-art (SOTA) on the standard 10-second benchmark, thanks to the novel architecture featuring token modulation and RoPE.

3. **Training cost**. FleXounDiT outperforms SOTA models with a significantly small model size. Moreover, the flexibility in the inference duration allows to train a TTA model with audio shorter than the target duration, which eliminates memory overhead during training, benefiting ones limited by low computational resources.

## 2 RELATED WORK

**Diffusion-based Text-to-Audio (TTA) models.** In TTA, a general framework is to use diffusion models pretrained on a large scale audio dataset. A seminal work is AudioLDM (Liu et al., 2023a;b) that introduces latent diffusion model framework adapted to audio data. The pipeline follows stable diffusion (Rombach et al., 2021) in image domain. In AudioLDM, the basic model consists of Variational Auto Encoder (VAE) and Latent Diffusion model. The latent diffusion model architecture is based on U-Net (Ronneberger et al., 2015) with convolutional neural networks (CNNs) as the main backbone. The extended version so-called AudioLDM2 (Liu et al., 2023b) allows to translate human-understandable text and audio representation from AudioMAE (Huang et al., 2022) into a single conditional framework. Another model is Tango (Ghosal et al., 2023) and Tango2 (Majumder et al., 2024) that adopts a similar framework as in AudioLDM. However, Tango2 uses a filtered dataset with better correspondences between text and audio. Thus, the representations from a text encoder (*e.g.*, T5 (Raffel et al., 2020)) better aligns text embeddings with sound events. In adopting diffusion

models for TTA, Make-an-audio (Huang et al., 2023b) proposes a diffusion based model composed of a 2D VAE and a convolutional U-Net denoiser equipped with pseudo prompt enhancement that well-aligned with audio. Additionally, Make-an-Audio2 (Huang et al., 2023a) enhances the model by incorporating a transformer architecture with improved text processing, structuring captions to define the beginning, middle, and end of the audio.

**Variable-length audio generation.** The diffusion transformer model (Peebles & Xie, 2022; Zheng et al., 2024; Huang et al., 2023a), while more advanced than convolution based U-Nets, struggles to scale beyond the size covered during training. The limitation arises from reliance on positional embeddings. The positional embeddings used in standard transformer is absolute. This limitation prevents arbitrary modification of sizes while maintaining performance. In Make-an-Audio2 (Huang et al., 2023a), The model can generate shorter audio because it is trained on multi-length data with a maximum duration of 27-second samples. However, the learnable position embeddings in transformer are fixed during testing, preventing the model from handling longer positions than those *seen* in training. Unlike standard diffusion transformer, CNN based diffusion models (Liu et al., 2023b; Majumder et al., 2024; Liu et al., 2023a; Ghosal et al., 2023) do not suffer from the limitation to scale the audio length due to the limitation of absolute position embeddings, but we empirically show in Sec. 5 that CNN models cannot preserve a consistent TTA performance for variable lengths.

**TSTL scenarios using rotary position embedding extension.** The absolute position embeddings in transformer architectures reduce flexibility in generating longer contexts (Su et al., 2021; Peng et al., 2024; Chen et al., 2023). To extend the context length without additional finetuning, positional embeddings with relative distance is required. In natural language processing, Rotary Position Embedding (RoPE) (Su et al., 2021) introduces a method to encode a position using a rotation matrix while simultaneously incorporating explicit relative position dependencies into the self-attention mechanism. RoPE acts as a base to achieve flexible length generation when integrated with interpolation techniques (emozilla, 2023; Chen et al., 2023; Peng et al., 2024) in the testing stage. Position interpolation (Chen et al., 2023), a RoPE feature scaling method, adjusts the new sequence to fit within the original context window by scaling between the original and target context lengths. This method applies the same scale to every dimension of key and query vectors in the attention module. However, the same scale for every dimension is sub-optimal. Specifically, as described by Neural Tangent Kernel (NTK) theory (Tancik et al., 2020), high-frequency dimensions are more challenging to learn than low-frequency dimensions in deep learning. As a result, NTK-aware interpolation recommends adjusting dimensions by scaling high frequencies less and vice versa. To further adjust the dimension regarding to benefit from position interpolation and NTK-aware interpolation, YaRN (Peng et al., 2024; Wang et al., 2024) proposes to use NTK-by-parts (emozilla, 2023) to adjust the frequency for each dimension equipped with attention scaling. Our proposed method makes use of the RoPE feature scaling technique and adapt the techniques to audio domain.

## 3 PRELIMINARIES

### 3.1 DIFFUSION MODELS

We consider a latent diffusion pipeline for TTA (Majumder et al., 2024; Liu et al., 2023a; Huang et al., 2023a), given an audio sample $\hat{u}$ and a condition either text or audio $y$. To encode the audio in the form of mel-spectrogram $u \in \mathbb{R}^{c \times \mathcal{F}' \times \mathcal{T}'}$, the Variational Auto Encoder (VAE) is employed to compress the input embedding $u$ to $x_0 \in \mathbb{R}^{\hat{D} \times \mathcal{F} \times \mathcal{T}}$ using an Encoder, and the decoder maps from the latent to the input space. The diffusion process gradually applies noise in the forward process $q(x_t|x_0)$ and reverse process $q(x_t|x_s)$ as follows:

$$q(x_t|x_0) = \mathcal{N}(x_t; \alpha_t x_0, \sigma_t^2 \mathbf{I}), \quad q(x_t|x_s) = \mathcal{N}(x_t; (\alpha_t/\alpha_s), \sigma_{t|s}^2 \mathbf{I}), \quad (1)$$

where $s < t$ and $\sigma_{t|s}^2$ is a scaled $\sigma_t^2$. Using reparameterization trick, we sample:

$$x_t = \alpha_t x_0 + \sigma_t \epsilon_t, \quad (2)$$

where $\epsilon_t$ is a Gaussian noise at timestep $t$. For the loss function, we define the target as v-prediction (Salimans & Ho, 2022) contrasting our approach from the previous TTA models (Liu et al., 2023b; Huang et al., 2023a; Majumder et al., 2024) which depend on $\epsilon$ as the target. In this approach, we define $\hat{v} = \alpha_t \epsilon - \sigma_t x_0$, and the loss functions is defined $\mathcal{L}_D = \|\hat{v} - v(x_t, t, y)\|^2$.

## 3.2 POSITION EMBEDDING

**Sinusoidal position embedding.** Position embedding (PE) plays an important role in transformer-based models to indicate the positions of the tokens within a specified sequence $L$. Sinusoidal PE falls into the absolute PE which is parameter-free and non-flexible in testing for longer sequences. Given a position on an axis $m$ object within an input sequence $0 \leq m \leq L/2$, the embeddings at $i$-th dimension use sinusoidal functions expressed as:

$$\boldsymbol{P}(m, 2i) = \sin\left(\frac{m}{b^{\frac{2i}{D}}}\right), \ \boldsymbol{P}(m, 2i+1) = \cos\left(\frac{m}{b^{\frac{2i+1}{D}}}\right), \tag{3}$$

where $D$ is the dimension of embeddings and $b$ is the base set to $10^4$. For simplicity, we denote $\boldsymbol{P}(m) \in \mathbb{R}^D$ ommitting the dimension index. Since this approach is parameter-free, sinusoidal embedding allows to extend the context length uncovered during training. The 2D extension of this approach is by applying a concatenation of the position based on $M = (m_x, m_y)$. The 2D sinusoidal embedding concatenates 1D sinusoidal embedding horizontally and vertically, and this serves as the foundational positional embedding in DiT (Peebles & Xie, 2022).

**Rotary Position Embedding (RoPE).** Su *et al*. (Su et al., 2021) introduce relative position embeddings, showing a capability of extending the context length in Large Language Models (LLMs). RoPEs incorporate the multiplication of Euler's formula $e^{i\theta}$ applied to key and query vectors. Given the $m$-th query $\boldsymbol{z}_m \in \mathbb{R}^D$ and $n$-th key $\boldsymbol{z}_n \in \mathbb{R}^D$, 1-D RoPE is formulated as:

$$f_q\big(\boldsymbol{z}_m, m, \boldsymbol{h}(\boldsymbol{\theta})\big) = \boldsymbol{W}_q \boldsymbol{z}_m e^{im\theta}, \ \ f_k\big(\boldsymbol{z}_n, n, \boldsymbol{h}(\boldsymbol{\theta})\big) = \boldsymbol{W}_k \boldsymbol{z}_n e^{in\theta}, \tag{4}$$

where $\theta$ is a rotary frequency value with a rotary base $b = 10^4$ and $\theta_i = b^{-\frac{2i}{D}}, i \in [1, 2, \cdots, D/2]$. The similarity score between relative distance $m - n$ between two embeddings can be formulated as:

$$g(\boldsymbol{z}_m, \boldsymbol{z}_n, m - n) = \text{Re}\Big(f_q\big(\boldsymbol{z}_m, m, \boldsymbol{h}(\boldsymbol{\theta})\big)^\top f_k\big(\boldsymbol{z}_n, n, \boldsymbol{h}(\boldsymbol{\theta})\big)\Big), \tag{5}$$

where $\boldsymbol{h}(\boldsymbol{\theta})$ is diagonal matrix of $\boldsymbol{\theta}$. Given a $m$-th token $\boldsymbol{z}_m$, the general form for matrix multiplication can be rewritten as:

$$f_{\{q,k\}}(\boldsymbol{z}_m, m, \boldsymbol{h}(\boldsymbol{\theta})) = \underbrace{\begin{bmatrix} \cos m\theta_1 & -\sin m\theta_1 & \dots & 0 & 0 \\ \sin m\theta_1 & \cos m\theta_1 & \dots & 0 & 0 \\ \vdots & \vdots & \ddots & \vdots & \vdots \\ 0 & 0 & \dots & \cos m\theta_{D/2} & -\sin m\theta_{D/2} \\ 0 & 0 & \dots & \sin m\theta_{D/2} & \cos m\theta_{D/2} \end{bmatrix}}_{\boldsymbol{R}_{\Theta_D}^m} \boldsymbol{W}_{\{q,k\}} \boldsymbol{z}_m.$$

$$\tag{6}$$

Furthermore, the token is encoded into a frequency-based embeddings depending on the rotary frequency $\theta$ and the position $m$.

## 4 FLEXOUNDIT - FLEXIBLE SOUND DIFFUSION TRANSFORMER

**Overview.** In this section, we present a novel method for variable-length audio generation. Following the well-established framework of latent diffusion models for audio generation (Huang et al., 2023b; Liu et al., 2023a;b; Ghosal et al., 2023; Majumder et al., 2024; Comunità et al., 2024), the audio Mel-spectrograms are encoded to a latent space by a 2D VAE, and can be restored to raw waveforms via a pretrained vocoder. We introduce a novel DiT block for TSTL scenarios, using our proposed frequency-based RoPE. To extend the lengths, we propose query-key scaling with Resonance YaRN (Wang et al., 2024) and frequency based attention scaling without test-time tuning. Fig. 1 illustrates our pipeline.

### 4.1 VARIABLE-LENGTH DIFFUSION TRANSFORMER ARCHITECTURE

**Flexible DiT block architecture.** Due to the lack of support for flexible generation in the DiT architecture (Peebles & Xie, 2022), we replace the standard attention to a RoPE attention module for enhanced position encoding in TSTL scenarios. Additionally, we apply LlamaRMSNorm (Touvron

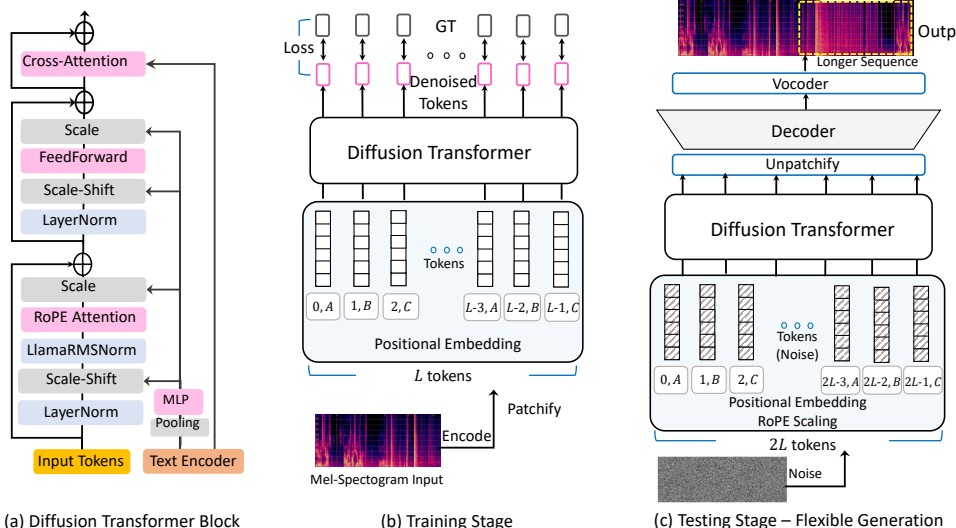

(a) Diffusion Transformer Block   (b) Training Stage   (c) Testing Stage – Flexible Generation

Figure 1: The illustration of the diffusion transformer block and pipeline of our proposed approach for training and testing. The diffusion transformer processes tokens with corresponding positions $x, y$ representing an index of a token position and an associated frequency location. The training stage only process up to $L$ sequence corresponding to a maximum audio length. The testing stage could process longer token sequences to produce longer audio lengths.

et al., 2023) to normalize the key and query embeddings, incorporating learnable parameters which is proven effective in LLAMA2 (Touvron et al., 2023). We also employ a text encoder (e.g., FLAN-T5 (Chung et al., 2024)) providing text embeddings, which are averaged and modulated via an MLP layer using the shift and scale as in FiLM (Perez et al., 2018). In addition, we introduce a cross-attention module to align audio generation with semantic content. Fig. 1 (a) illustrates our proposed transformer block for the TSTL scenario.

**Patchification.** Given the Mel-spectrogram encoded by the VAE, we obtain a latent embedding $\boldsymbol{x} \in \mathbb{R}^{\hat{D} \times \mathcal{F} \times \mathcal{T}}$, which is further patchified with a patch size of $p \times p$ (Koutini et al., 2021; Niizumi et al., 2022; Huang et al., 2022; Zhong et al., 2023). This transforms the latent input into a 2D sequence of size $\mathbb{R}^{D \times F \times T}$, where $F = \frac{\mathcal{F}}{p}, T = \frac{\mathcal{T}}{p}$. The patchification converts the 2D latent inputs along the frequency and time axes to tokens as illustrated in Fig. 2 (left). This patchification downsamples the input latents by $p^2$ times into $L = F \times T$ tokens. Then, we obtain a sequence $\boldsymbol{z} \in \mathbb{R}^{L \times D}$ as the input to the transformer model.

**Frequency-based RoPE.** Even though 2D RoPE effectively handles positional embeddings in images, applying RoPE directly to the Mel-spectrograms is problematic due to the harmonic structures along the frequency axis. Specifically, a longer sound event is encoded to the latent space with a fixed number of tokens in the frequency axis, despite more tokens are produced in the time axis. To preserve this structure, we propose using sinusoidal PEs for the frequency axis to equip the embeddings under one temporal frame with absolute positions. To allow audio generation with flexible duration, we incorporate 1D RoPE to build the base of the attention module with the token sequence. Consequently, a DiT equipped with the proposed frequency-based RoPE benefits from both the absolute and relative distance to better model the structure of the latent Mel-spectrogram space. Let a $m$-th query $\boldsymbol{q}_m = f_q(\boldsymbol{z}_m, m, \boldsymbol{h}(\boldsymbol{\theta}))$ and a $n$-th key $\boldsymbol{k}_n = f_k(\boldsymbol{z}_n, n, \boldsymbol{h}(\boldsymbol{\theta}))$, the attention score in RoPE can be formulated as:

$$A_{m,n} = \text{softmax}\Big(\frac{\langle \boldsymbol{q}_m, \boldsymbol{k}_n \rangle}{\sqrt{D}}\Big). \tag{7}$$

Given a mapping function $\phi$, the absolute PE is integrated to the formulation yielding the similarity between the query $\boldsymbol{q}_m$ and key $\boldsymbol{k}_n$ as stated below:

$$\langle \boldsymbol{q}_m, \boldsymbol{k}_n \rangle = \phi(\boldsymbol{z}_m + \boldsymbol{P}(m))^\top \boldsymbol{W}_q^\top (\boldsymbol{R}_{\theta_D}^m)^\top \boldsymbol{R}_{\theta_D}^n \boldsymbol{W}_k \phi(\boldsymbol{z}_n + \boldsymbol{P}(n)). \tag{8}$$

For each $\theta_i$, the wavelength $\lambda_i$ of the RoPE embedding at the $i$-th hidden dimension is defined as the token length for a complete rotation, expressed by $\lambda_i = 2\pi b^{\frac{2i}{D}}$.

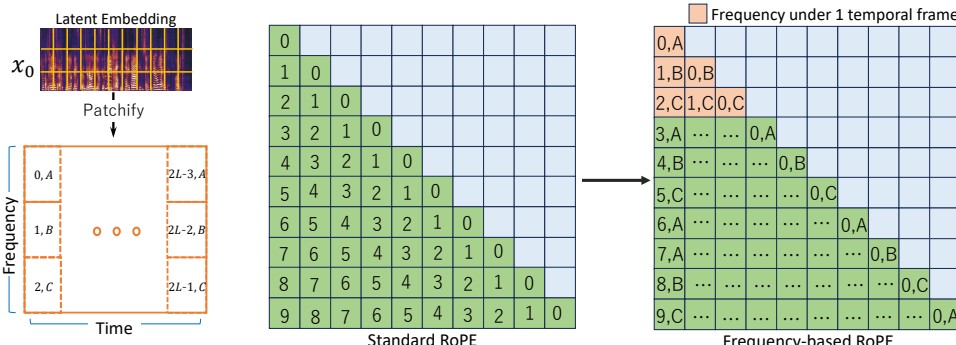

Figure 2: An illustration of patchification and the occurrence map impacted by the choice of PEs. (Left) The latent input is tokenized along the time axis, encoding frequency with absolute PEs combined with RoPEs. (Middle) A Toeplitz matrix illustrates position indexes in RoPE. (Right) The matrix is reformed to reflect relative positions of tokens across various frequencies in a time frame.

### 4.2 TUNING-FREE VARIABLE-LENGTH GENERATION

**Training and testing strategies for variable-length generation.** To achieve variable-length generation, we need to address the problems of testing on the lengths uncovered during training (TSTL) and the lengths covered during training (TTCL). To this end, we propose two strategies: 1) a novel RoPE feature interpolation framework with a frequency-based scaling term, and 2) training on varying lengths. Note that, we apply the RoPE feature interpolation only in the TSTL scenario where the target length in testing is longer $L' > L$. We define our contribution in extending RoPE features applied to the audio domain below.

**How to test on longer audio when trained on shorter samples?** A way to generate the audio in TSTL is by extending the token sequence from $L$ to $L'$, where $L' > L$. To work on a longer token sequence, we require modification to the base $b$ of the rotary frequency $\boldsymbol{h}(\boldsymbol{\theta}) = \text{Diag}(\theta_1, \cdots, \theta_{D/2})$, also known an interpolation technique for RoPE feature extension. One strategy to scale the base $b$ is using YaRN (Peng et al., 2024). Instead of arbitrarily changing the base $b$, *NTK-by-parts* is applied for targeted interpolation on each dimension. In other words, the base is modified based on the dimension by constructing a piecewise linear function as follows:

$$h(\theta_i) = \left(1 - \gamma(r_i)\right)\frac{\theta_i}{s} + \gamma(r_i)\theta_i, \tag{9}$$

where a ramp up function is denoted as $\gamma$ and a scaling ratio between the target and training lengths is denoted as $s = \max(\frac{L'}{L}, 1.0)$. Because YaRN depends on the wavelength $\lambda_i$ to define the rotary frequency for each $i$-th dimension, we use the ramp up function as follows:

$$\gamma(r_i) = \begin{cases} 0, & \text{if } r_i < \alpha \\ 1, & \text{if } r_i > \beta \\ \frac{r_i - \alpha}{\beta - \alpha}, & \text{otherwise.} \end{cases} \tag{10}$$

Here, $\alpha, \beta$ are hyperparameters (see details in App. A). To linearly interpolate by scale $s$, the ratio $r$ is computed depending on $i$ as follows:

$$r_i = \frac{L}{\lambda_i} = \frac{L}{2\pi b^{\frac{2i}{D}}}. \tag{11}$$

In essence, the *NTK-by-parts* approach only focuses on the condition where $\lambda_i \geq L$. which is shown problematic in (Wang et al., 2024). As the position index $m$ increases, a phase shift occurs for the rotary angle after each full rotation. To mitigate this issue, we apply the Resonance YaRN technique (Wang et al., 2024) to adjust the rotary frequency for $\lambda_i < L$, expressed as:

$$\hat{\theta}_i = \frac{2\pi}{\hat{\lambda}_i}, \tag{12}$$

where the wavelength is rounded to the closer integer value $\hat{\lambda}_i = \text{round}(\lambda_i)$. The frequency scale of RoPE for each $i$-th dimension is updated to $\hat{\theta}_i$.

**Frequency-based dynamic attention scaling technique.** The Attention scaling technique with a scale $\mu$ is critical when RoPE is expanded for longer sequence generation (Peng et al., 2024; Wang et al., 2024; Su, 2023) . The scale $\mu$ to enhance length *extrapolation* capabilities is applied for the query and key, which serves as a temperature to the attention operation:

$$\boldsymbol{q}'_m = \mu \cdot \boldsymbol{q}_m, \quad \boldsymbol{k}'_n = \mu \cdot \boldsymbol{k}_n. \tag{13}$$

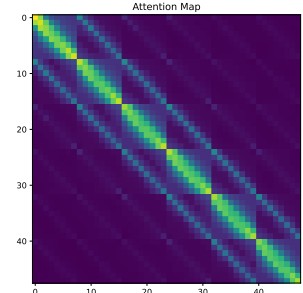

Attention Map

In YaRN (Peng et al., 2024), the scale of RoPE attention map is presented by a constant factor $\mu = 0.1 \ln(s) + 1$. However, this might be sub-optimal compared to dynamic scaling as discussed in (Zhang et al., 2024b). In addition to a vanilla dynamic scaling, we believe that the attention scale must follow the structure of the frequency interval $F$ in the token sequence to preserve the attention *blocks* intact when expanded for longer sequence. This idea is based on our observation that the attention map forms several active *blocks* corresponding to frequency within a single temporal frame in Fig. 3. To this end, we define our proposed attention scaling factor as stated below:

Figure 3: An illustration shows the pattern in the attention map forms *active* attention blocks corresponding the frequency tokens under a temporal frame.

$$\mu = \frac{\log\left(F \cdot \text{round}(\frac{m}{F}) + 1\right)}{\log(L)}. \tag{14}$$

Note that we clip the minimum value using the scaling factor $\mu = 0.1 \ln(s) + 1$.

**Training on varying lengths.** While the above proposed resonance YaRN and the frequency-based dynamic attention scaling well addresses the challenge of generating sound events with an unseen long duration *i.e.*, TSCL, we observe that training exclusively on a fixed audio length limits the ability to generate audio *shorter* than the trained length. To ensure the robustness in handling various *shorter* durations, we crop the 10-sec audio clip to 2.5-sec, 5-sec, 7.5-sec or 10-sec (no cropping) by a uniform distribution during training. This augmentation is simple to implement and is beneficial for memory usage. For details, please see App. B.

## 5 EXPERIMENTS

**Overview.** In experiments, we perform evaluation on text to audio generation on AudioCaps (Kim et al., 2019) and Clotho (Drossos et al., 2020) datasets. We compare with SOTAs and reevaluate the results of recent methods on our settings. Moreover, we present the capability of our technique in inpainting and outpainting of the audio. We also present analysis to the proposed method to delve into details of each component. Besides comparison by objective metrics, we also provide subjective evaluation that demonstrates the efficacy of our proposed approach.

### 5.1 IMPLEMENTATION

**Datasets.** The datasets used in this work include WavCaps (Mei et al., 2023), Audiocaps (Kim et al., 2019), and Clotho (Drossos et al., 2020). WavCaps is a dataset containing 400K audio clips with weakly-labeled captions generated with assistance from ChatGPT. AudioCaps, a subset of AudioSet (Gemmeke et al., 2017), comprises approximately 46K 10-second clips with manually annotated captions. All these datasets are converted into 10 seconds with 16kHz. For evaluation on AudioCaps, we only focus on training and testing on the same dataset. Furthermore, for evaluation on Clotho, we train our model on WavCaps and Clotho-train then test on Clotho-eval.

**Objective and subjective evaluation.** Our primary evaluation is on the TTA generation task to assess the quality performance of generated samples. We adhere to the evaluation protocol of past works (Kreuk et al., 2022; Liu et al., 2023b; Huang et al., 2023a), which involves calculating objective metrics *e.g.*, Frechet Audio Distance (FAD) and Kullback-Leibler divergence (KL). For KL divergence, we make use of the pretrained PaSST model (Koutini et al., 2021). Besides quality-based metrics, we also compare alignment between generated audio to the corresponding text using the LAION-CLAP score (Wu et al., 2023). Across experiments, we observe that FAD scores is a realiable measurement to assess objectively for variable-length generation. For FAD evaluation on variable-length audio generation, we compare the distribution of the generated variable-length audio

Table 1: Evaluation results and comparison with state-of-the-arts on AudioCaps. † indicates that we re-evaluate the models using publicly available pretrained models and remove the sample selection stage with high CLAP scores in AudioLDM2 and Tango2.

| Model | Param. | Text Cond. | FAD (↓) | KL (↓) | CLAP (↑) | OVL (↑) | REL (↑) |
|---|---|---|---|---|---|---|---|
| AudioLDM-Large (Liu et al., 2023a) | 739M | CLAP | 1.96 | 1.59 | - | - | - |
| Make an audio (Huang et al., 2023a) | 453M | CLAP | 2.66 | 1.61 | 0.21 | - | - |
| TANGO-AC (Ghosal et al., 2023) | 866M | FLAN-T5 | 1.73 | 1.27 | 0.19 | - | - |
| AudioLDM2-AC-Large (Liu et al., 2023b) | 1.5B | FLAN-T5 | 1.42 | **0.98** | 0.24 | - | - |
| AudioLDM2-Full-Large† (Liu et al., 2023b) | 1.5B | FLAN-T5 | 3.20 | 1.73 | 0.22 | 3.12 | 2.89 |
| TANGO2-Full† (Majumder et al., 2024) | 1.20B | FLAN-T5 | 2.41 | 1.22 | **0.25** | 3.14 | 3.65 |
| Make-an-Audio2† (Huang et al., 2023a) | 937M | T5 + CLAP | 1.33 | 1.24 | 0.24 | 3.32 | 3.51 |
| SA-Open† (Evans et al., 2024b) | 1.30B | T5 | 3.77 | 2.30 | 0.19 | 2.82 | 2.16 |
| FleXounDiT (ours) | 612M | FLAN-T5 | **1.24** | 1.45 | **0.25** | **3.38** | **3.92** |

with that of the target 10-second audio. We also conduct human assessments as subjective evaluation to measure audio quality and the text-audio alignment fidelity. The test consists of two questions about OVeralL impression (OVL) and text-audio RELevance (REL) with 5 points, where 1 and 5 indicate poor and excellent quality, respectively.

**Training and testing details.** For reproducibility, we employ the publicly available VAE (Liu et al., 2023a;b; Ghosal et al., 2023) to compress the Mel-spectrogram into a latent representation with a downsampling rate of 4 at the sampling rate of 16kHz. Our diffusion model is trained on 8 NVIDIA A100 GPUs, using 110K optimization steps and a batch size of 32 per GPU. We employ the AdamW optimizer (Loshchilov & Hutter, 2019) with a learning rate of 1.5e-4. For vocoder, we utilize the HiFiGAN (Kong et al., 2020) released by AudioLDM (Liu et al., 2023a). Further, FLAN-T5 (Chung et al., 2024) is used as the text encoder. We train on various lengths ∼10 seconds for comparison with prior TTA models and tests are performed on the base audio of 10 seconds unless otherwise specified. In testing, our model is evaluated with a classifier-free guidance scale of 3.5. See details in App. B.

**Comparison with SOTAs.** For evaluation, we compare with SOTAs in TTA generation. In this experiment, we particularly pick recent TTA diffusion models (*e.g.*, AudioLDM2 (Liu et al., 2023b), Tango2 (Majumder et al., 2024), and Make-an-Audio2 (Huang et al., 2023a)) and re-evaluate these models for TTA tasks. Note that, we include Stable Audio Open (SA-Open) (Evans et al., 2024b) in comparison for completeness. We realize that the comparison might not be fair as the model focuses on stereo audio with 44.1kHz, while our proposed approach and the other prior works associated with 16kHz mono audio. Nevertheless, we provide subjective results for the recent models to assess overall impression and fidelity.

## 5.2 RESULTS

**Standard length TTA generation.** Evaluation in this experiment is a common benchmark testing on 10-second audio. Table 1 and 2 demonstrate the effectiveness of our method, outperforming previous approaches on the AudioCaps and Clotho datasets. Our proposed method surpasses the SOTAs with lower FAD scores and higher CLAP scores, while also requiring fewer parameters for training compared to prior works. Our work also demonstrates superiority over SOTA methods in subjective listening tests indicated by OVL and REL scores. Notably, our approach achieves high performance in terms of text-audio relevance scores, while we observe that the recent TTA models perform worse, especially in generating audio that accurately aligns with the text description consisting of multiple sounds. Please see our demonstration page[1] to listen the generated audio and compare with SOTAs.

**Variable-length TTA generation.** For variable-length generation shown in Fig. 4 (a), our method preserves the FAD scores below 2.0 across different durations. Other prior TTA models fall short, experiencing performance degradation when generating the audio for longer durations which are not well-covered during training. Fig. 4 (a) demonstrates the robustness of our method against varying

---

[1] https://flexoundit.github.io/

Table 2: Evaluation results and comparison with SOTAs on the Clotho dataset. The Clotho-eval set is used for testing.

| Model | FAD (↓) | KL (↓) |
|---|---|---|
| TANGO (Ghosal et al., 2023) | 3.61 | 2.59 |
| AudioLDM (Liu et al., 2023a) | 4.93 | 2.60 |
| Make-An-Audio2 (Huang et al., 2023a) | 2.13 | 2.49 |
| FleXounDiT (ours) | **1.95** | **2.21** |

Table 3: Comparison with DiT using Absolute PE and RoPE on AudioCaps. The models are only trained on 10-second audio and tested on 20-second and 30-second audio.

| Model | PE | FAD (↓) | | |
|---|---|---|---|---|
| | | 10 sec. | 20 sec. | 30 sec. |
| DiT (Peebles & Xie, 2022) | RoPE - No Ext. | 3.20 | - | - |
| DiT (Peebles & Xie, 2022) | Absolute PE | 1.89 | 3.93 | 5.50 |
| FleXounDiT (ours) | Freq. based RoPE | **1.32** | **1.67** | **1.60** |

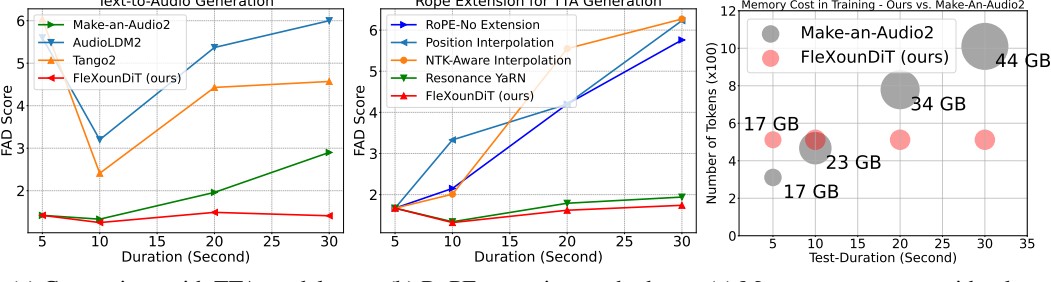

(a) Comparison with TTA models    (b) RoPE extension methods    (c) Memory cost assoc. with tokens

Figure 4: (a) A comparison of current TTAs for generating variable-length audio ranging from 5 to 30 seconds reveals that the performance of audio generation declines as the duration increases. (b) A comparison of current length extension adopted from NLP. The model is pretrained on 5-second audio. The Position Interpolation and NTK-aware interpolation cannot handle length extension properly in audio data. (c) Efficiency in the training short test long scenario between ours vs. Make-an-Audio2. We measure in terms of the number of processing tokens and the memory cost on GPUs.

time-lengths compared to SOTAs. The U-Net-based models, *e.g.*, AudioLDM2 (Liu et al., 2023b) and Tango2 (Majumder et al., 2024), which are exclusively trained on 10-second audio, drop over 2 points in the FAD performance on variable-length generation. In addition, Make-an-Audio2 (Huang et al., 2023a) also shows a decline in performance when the audio lengths are extended beyond 10 seconds. It is worth mentioning that there are no additional time conditions or post processing (*e.g.*, truncating audio) applied to our model and outputs. This indicates that our method is easy to use.

**Comparison with other RoPE extension techniques.** In this experiment, we compare with the past techniques for RoPE extension for longer duration generation. To evaluate the RoPE extension against the SOTAs, we opt to compare with (1) no RoPE extension, (2) Position Interpolation (Chen et al., 2023), (3) NTK-aware interpolation (bloc97, 2023), and (4) vanilla Resonance YaRN (Wang et al., 2024) with our proposed method. We train on 5-second audio and extend the lengths to audio longer than 5 seconds. We can observe in Fig. 4 (b) that methods without attention scaling techniques (*e.g.*, NTK-aware interpolation and Position Interpolation) are not effective to extend to unseen long sequence in our audio generation framework. Also, our proposed method with frequency based attention scaling technique outperforms the performance of the Resonance YaRN technique (Wang et al., 2024) with a constant scale.

## 5.3 ANALYSIS AND ABLATION STUDIES

**Frequency based RoPE vs. Standard RoPE.** We observe that simply replacing all absolute PEs in DiT to RoPEs is deteriorating the performance. Table 3 shows the FAD performance on 10-sec AudioCaps drops from 1.89 (Absolute PE) to 3.20 (vanilla RoPE). We conjecture that the tokens associated to the frequency axis should be encoded to an absolute position embedding to preserve the structure of the Mel-spectrograms. While, using frequency based RoPE could achieve a better FAD of 1.32 compared to DiT with Absolute PE (FAD=1.89) when trained solely on 10-second audio. We observe that implementing DiT by directly substituting absolute PE with RoPE is inadequate, and since the model is not well-converged, we only perform comparisons using 10-second audio.

Table 4: Impact of attention scaling. The model is trained on 5-second audio then extending up to 30 seconds. The base RoPE extension is Resonance YaRN.

| FleXounDiT - Attention Scale | FAD (↓) | | | |
|---|---|---|---|---|
| | 5 sec. | 10 sec. | 20 sec. | 30 sec. |
| - W/O scale | 1.67 | 2.40 | 4.79 | 5.41 |
| - W/ constant scale | 1.67 | 1.37 | 1.75 | 1.97 |
| - W/ freq. based dynamic scaling | 1.67 | **1.32** | **1.65** | **1.79** |

Table 5: Evaluation on inpainting and outpainting for the standard and extended lengths. The upperbound is based on VAE reconstruction of the original inputs.

| | 10 sec. | | 20 sec. | |
|---|---|---|---|---|
| | FAD (↓) | KL (↓) | FAD (↓) | KL (↓) |
| VAE Recon. (Upperb.) | 0.94 | 0.34 | 0.94 | 0.34 |
| Inpainting | 1.10 | 0.69 | 1.27 | 1.00 |
| Outpainting | 0.95 | 0.58 | 1.23 | 0.85 |

**Frequency based RoPE vs. Absolute position embedding.** DiT (Peebles & Xie, 2022) is a natural extension of existing diffusion models *e.g.*, AudioLDM (Liu et al., 2023a) and Tango (Majumder et al., 2024) to improve performance. We compare with DiT with absolute PEs, namely sinusoidal PEs, and without RoPE embeddings. We train the DiT model on AudioCaps and then apply extrapolation to the PEs in a pretrained DiT. The standard DiT cannot generate audio following the text description after 10 seconds (see qualitative comparison in App. B). Table 3 shows the superiority of our proposed method on AudioCaps in generating variable-length audio compared to DiT with absolute PE.

**Cost efficiency in the TSTL scenario.** Another impact of our proposed method is the ability to reduce training cost. Especially, we can train on a short sequence that requires only low memory overhead then generating the sample with longer sequence. As shown in Fig. 4 (c), Make-an-Audio2 needs to be trained on the longer sequence to perform on the specified longer sequence in testing. While, our proposed method could preserve the number of processing tokens during training to perform on longer sequences. This observation certainly reduces the need for training cost on longer audio sequences to work on the specified longer durations. In terms of resource usage, Make-an-Audio2 (Huang et al., 2023a) requires an additional ∼10GB of GPU memory to handle the increased number of tokens when extending audio generation by 10 seconds. Our proposed method could avoid additional memory requirements by learning from shorter audio.

**Impact of attention scaling.** In Table 4, We can observe that despite the RoPE and frequency adjustment applied to the model, addressing TSTL is not effective without the attention scaling technique. The RoPE scaling method that employs a constant factor, as demonstrated in Resonance YaRN (Wang et al., 2024), is promising in extending sequence lengths to a certain extent, but it might not fully optimize performance across all settings. While, our proposed frequency-based dynamic scaling approach exhibits improvements. As shown in Table 4, our attention scaling technique outperforms both the constant scaling method (*e.g.*, Resonance YaRN (Wang et al., 2024)) and without scaling, thereby validating the effectiveness of our approach.

## 5.4 FINETUNING-FREE AUDIO INPAINTING AND OUTPAINTING TASKS

In this experiment, we demonstrate another capability of our proposed method in inpainting and outpainting for masked audio inputs. In painting and outpainting tasks are performed on 10-second and 20-second audio. In the inpainting task, we mask the inputs for 15 seconds in the middle with 2.5 second for each end. In the outpainting task, we mask the inputs for 17.5 seconds starting from the timestamp 2.5 seconds. To test the capability on these tasks, we employ the AudioCaps test set. A pretrained FleXounDiT performs on this task without further finetuning. Table 5 shows that our method can preserve the FAD and KL scores low against the upperbound scores which are obtained from VAE reconstruction of the original inputs.

## 6 CONCLUSIONS

In conclusion, our proposed text-to-audio (TTA) diffusion model so-called FleXounDiT appears to address the challenge of generating variable-length audio, a limitation in existing TTA diffusion models. By using a novel framework based on relative positional embeddings, our proposed approach enables the generation of audio at arbitrary lengths without requiring additional training or input conditions, thereby improving computational efficiency. This training-free approach not only reduces resource demands but also delivers high performance for various TTA tasks, demonstrated by surpassing state-of-the-art methods in both standard audio generation benchmarks and variable-length audio generation, while utilizing a significantly smaller model size.

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

## A  MODEL ARCHITECTURE DETAILS

**Our architecture and design details.**   We set the depth of transformer blocks to 12 with a token hidden size set to 768. The number of attention head is set to 16. The latent input for DiT is in the size of $8 \times 256 \times 16$. For patchification, we use a 2D convolution layer with a kernel size of 2 (*i.e.*, patch size $p = 2$) and transform the embedding size to 768. We use FLAN-T5 (Chung et al., 2024) to encode text and use all embeddings out of this model to pair with audio embeddings from our DiT in cross-attention modules. To encode the text for shift and scale modules, we use a simple average pooling operation to aggregate the text embeddings.

**VAE and vocoder options.**   We developed FleXounDiT upon the VAE and vocoder (*i.e.*, the reconstruction pipeline) used in AudioLDM series (Liu et al., 2023a;b) for two reasons. (1) *Model size*. The reconstruction pipeline in Make-an-Audio2 (Huang et al., 2023a) has a similar performance to the AudioLDM pipeline in terms of FAD score (around 1.0 on AudioCaps) (Huang et al., 2023a). In terms of the model size, the reconstruction pipeline of Make-an-Audio2 has a larger size of ∼215M parameters for VAE and ∼100M for the BigVGAN vocoder (Lee et al., 2022), while the AudioLDM VAE only comprises of ∼115M parameters, with a HiFiGAN vocoder of ∼15M parameters. It is also observed that the light-weight HiFiGAN is not the bottleneck for the Mel-spectrogram reconstruction pipeline (Comunità et al., 2024). (2) *Wide adoption*. The AudioLDM reconstruction pipeline has been widely adopted in recent TTA research (Ghosal et al., 2023; Majumder et al., 2024; Saito et al., 2024). By following the identical reconstruction pipeline, we concentrate on the design of the diffusion denoiser that is flexible with variable lengths.

**On the adoption of RoPE in SOTA transformer models.**   The transformer module is adopted in Make-an-audio2 (Huang et al., 2023a). However, the module only consists of cross-attention to the input conditions. This design prohibits the use of RoPE which is implemented for self-attention modules. RoPE has been adopted for sound source separation (Lu et al., 2024), and recently introduced to audio generation tasks (Liu et al., 2024; Evans et al., 2024a;b) for the sake of performance. Nevertheless, none of the aforementioned transformer architectures equipped with RoPE is designed to work on generating longer audio lengths beyond what are covered during training (TSTL), which differentiates FleXounDiT from prior arts.

**RoPE extension.**   In our proposed approach, we make use of RoPE extension derived from Resonance YaRN (Peng et al., 2024; Wang et al., 2024). We need to set the hyperparameters to define the ramp up function Eq. (10) as described in the main paper. These hyperparameters are to define interpolation to RoPE features. Recall in Eq. (10), all hidden dimensions $i$ where $r_i < \alpha$ are linearly interpolated by a scale $s$, and the dimensions where $r_i > \beta$ are not interpolated. In our implementation, we set $\alpha = 1$ and $\beta = 32$. As the number of processing tokens for 10-second sound event in our model is 1024, we calculate the scale $s$ based on this number.

## B  DETAILS AND ADDITIONAL EXPERIMENTS

**Implementation and experiment details.**   To complement the implementation and experiment details of our main page, we provide additional information below. The first stage in latent diffusion process is to train the VAE model. In this case, we make use off-the-shelf VAE model as well as the HiFiGAN vocoder provided by AudioLDM series (Liu et al., 2023a;b). As we target to generate audio from text description, we could use any text encoder (*e.g.*, CLAP (Wu et al., 2023) or T5 (Raffel et al., 2020)). However, the CLAP model (Wu et al., 2023), which computes global-level conditions, has been observed to struggle with capturing temporal information in text data. To address this, we employ a different pretrained text encoder to capture the semantic details of the textual input, which may include important temporal sequences. Specifically, we utilize FLAN-T5 (Chung et al., 2024), an enhanced version of the text-to-text transfer transformer model (Raffel et al., 2020), which has been finetuned on a variety of tasks. Unlike some prior arts (Liu et al., 2023a; Comunità et al., 2024) that can train using audio only, our proposed method needs to be trained using text-audio pairs on the datasets. In addition, for analysis on memory usage, we use NVIDIA RTX A6000 with 32 batch size.

**Numerical details.**   In this section, we present numerical details of the task generating various audio lengths and compare our proposed method to SOTAs in Fig. 4 (a) of the main paper. In table 6, CNN

based models (*e.g.*, AudioLDM2 and Tango2) cannot generalize well on variable-length generation. Note that, we use off-the-shelf models of AudioLDM2 from the checkpoint "audioldm2-large" and Tango2 from the checkpoint "Tango-Full-FT-Audiocaps". In addition, the transformer-based model (*e.g.*, Make-an-Audio2) cannot generalize to the lengths out of which covered during training, which is around 27 seconds. Our proposed model consistently preserves the FAD performance below 2.0 across different audio durations. Furthermore, we also provide Table 7 to show details of RoPE extension corresponding to the Fig. 4 (b). We begin with a pretrained FleXounDiT model on 5-second audio Audiocaps and evaluate various RoPE extension methods. The 5-second audio is a trimmed version of 10-second audio on AudioCaps. We observe that the methods without attention scaling technique (*e.g.*, position interpolation (Chen et al., 2023), NTK-aware interpolation (bloc97, 2023), and no extension) could not preserve the FAD scores below 2.0 on AudioCaps, indicating the attention scale is vital for audio length extension. In contrast, the methods using attention scaling techniques (*e.g.*, Resonance YaRN (Wang et al., 2024) and FleXounDiT) does not suffer from significant performance degradation when extending the audio lengths.

Table 6: Evaluation results on variable-length audio generation. This evaluation is trained on various lengths with a maximum of 10-second audio and tested to generate variable-length audio.

| Model | FAD ($\downarrow$) | | | |
|---|---|---|---|---|
| | 5 secs | 10 secs | 20 secs | 30 secs |
| AudioLDM2-Full-Large | 5.60 | 3.20 | 5.37 | 6.35 |
| Make an audio2 | 1.42 | 1.33 | 1.95 | 2.90 |
| TANGO2-Full | 6.05 | 2.41 | 4.43 | 4.55 |
| FleXounDiT (ours) | **1.42** | **1.24** | **1.48** | **1.39** |

Table 7: Evaluation results on extending audio lengths using RoPE extension methods. This evaluation is trained on 5-second audio and tested on various extended lengths.

| Method | FAD ($\downarrow$) | | | |
|---|---|---|---|---|
| | 5 secs | 10 secs | 20 secs | 30 secs |
| RoPE - No Extension | 1.67 | 2.15 | 4.20 | 5.76 |
| Position Interpolation (Chen et al., 2023) | 1.67 | 3.33 | 4.2 | 6.23 |
| NTK-aware Interpolation (bloc97, 2023) | 1.67 | 2.01 | 5.55 | 6.27 |
| Resonance YaRN (Wang et al., 2024) | 1.67 | 1.37 | 1.75 | 1.97 |
| FleXounDiT (ours) | 1.67 | **1.32** | **1.65** | **1.79** |

**Resonance YaRN and vanilla YaRN.** As we discuss in the main paper, Resonance YaRN (Wang et al., 2024) shows the efficacy over a standard YaRN (Peng et al., 2024). In vanilla YaRN, the method only focuses on the frequency scaling to the dimension with the wavelength $\lambda_i \geq L$. The resonance YARN resolves this issue using a round to integer function to reduce frequency shifts when working on longer sequences. Resonance YaRN improves over vanilla YaRN for generating longer sequences and trained on 5-second audio. Table 8 presents the results on AudioCaps for various audio durations.

Table 8: Vanilla YaRN vs. Resonance YaRN with a constant attention temperature.

| Model | FAD ($\downarrow$) | | | |
|---|---|---|---|---|
| | 5 sec. | 10 sec. | 20 sec. | 30 sec. |
| Vanilla YaRN | 1.67 | 1.41 | 1.82 | 2.20 |
| Resonance YaRN | 1.67 | 1.37 | 1.75 | 1.97 |

**Impact of training on varying lengths.** In this section, we validate the impact of varying length training compared to only training the model with 5 second or 10 second audio clips. In Table 9, we observe that in the TSTL scenario, all three settings could generate longer audio lengths with preserved FAD scores (below 2.0), generating an unseen shorter length could be problematic as

shown in the "Only 10 sec. audio" setting that degraded the FAD score to 2.10 when generating 5-sec audio.

Table 9: Comparison of training only on a single audio length and various audio lengths. In fixed-length training, we only provide 5 and 10 second audio then testing to generate 5-30 second audio. We compare the fixed-length training with various-length audio training (*e.g.*, 2.5-10 seconds).

| Training Strategy | Testing - Generation | FAD ($\downarrow$) |
|---|---|---|
| **Only 5 sec. audio** | 5 sec. | 1.67 |
| | 10 sec. | 1.36 |
| | 20 sec. | 1.62 |
| | 30 sec. | 1.94 |
| **Only 10 sec. audio** | 5 sec. | 2.10 |
| | 10 sec. | 1.32 |
| | 20 sec. | 1.67 |
| | 30 sec. | 1.60 |
| **Various 2.5, 5, 7.5, 10 sec. audio** | 5 sec. | 1.42 |
| | 10 sec. | 1.24 |
| | 20 sec. | 1.49 |
| | 30 sec. | 1.39 |

**Qualitative results.** In this section, we provide qualitative results of our proposed method for generating variable-length audio, comparing with vanilla DiT, inpainting, and outpainting masked audio. Fig 5 shows our generated audio on various durations from 5 seconds to 20 seconds with the corresponding text description. Fig. 6 shows a comparison between our proposed approach with frequency based RoPE and the absolute PE with sinusoidal PE used by a standard DiT. We also show inpainting and outpainting results on 20-second audio in Fig. 7.

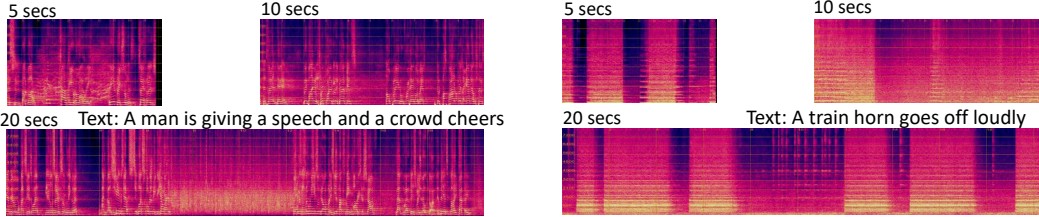

Figure 5: Qualitative results for generating various audio durations.

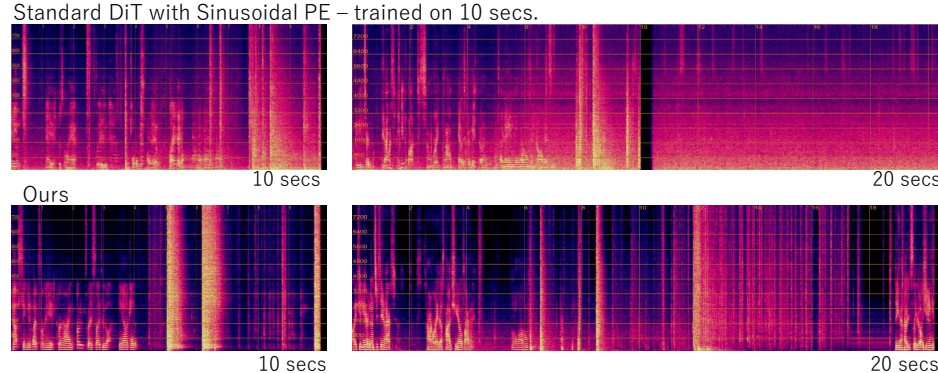

Figure 6: Qualitative results of our FleXounDiT with frequency based RoPEs vs. standard DiT with absolute PEs.

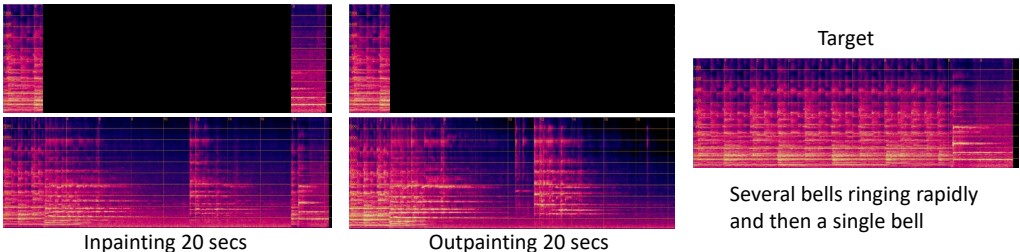

Target

Several bells ringing rapidly
and then a single bell

Inpainting 20 secs          Outpainting 20 secs

Figure 7: Qualitative results for inpainting and outpainting.

## C  LIMITATIONS AND BROADER IMPACTS

**Limitations.**   Our proposed method have some limitations when it comes to accurately matching audio generation with the intended semantic meanings. In certain cases, the model may generate sound effects for only a subset of the sounds mentioned in the text, even though the text may describe multiple different sounds. This results in incomplete outputs that do not fully capture the range of elements present in the input text description. Addressing this limitation requires further refinement in the text description and structure to map textual cues to corresponding audio components comprehensively.

**Broader impacts.**   From an ethical perspective, generative models operate probabilistically when producing sounds. This means the generated audio might sometimes be undesirable, particularly if the model is trained on inappropriate or unwanted data points.Thus. model deployment might also raise concerns, particularly regarding the potential for generating misleading contents and propagating biases that present in training data. This phenomenon might yield bad perception from humans to the text-to-audio models.

