# OpenReview forum: "FLEXOUNDIT: VARIABLE-LENGTH DIFFUSION TRANSFORMER FOR TEXT-TO-AUDIO GENERATION"
_ICLR.cc/2025/Conference — ICLR 2025 Conference Withdrawn Submission_

### Official Review · Reviewer_HuiJ · 2024-10-15

**Soundness:** 2
**Presentation:** 2
**Contribution:** 1
**Rating:** 3
**Confidence:** 5

**Summary:**

This work presents a framework using relative position embeddings to enable variable-length audio generation in text-to-audio diffusion models, addressing the challenges of extrapolation beyond training durations. The approach allows tuning-free audio length extrapolation, reduces training costs with shorter audio durations, and outperforms existing state-of-the-art methods in audio generation benchmarks while maintaining a smaller model size.

**Strengths:**

This paper aims to solve the text-to-audio task. The advantages includes:

(1) present a fair performance on public evaluation dataset.

(2) low training cost.

**Weaknesses:**

This paper raises several concerns, including:

(1) Lack of novelty:

 a. Generating variable-length audio is neither a difficult nor a novel problem.

 b. Using Rotary Position Embedding (RoPE) for position encoding to handle variable lengths is already widely known across different fields, such as LLMs and diffusion generative models. Moreover, nearly all generative models attempt to use Transformer backbones along with RoPE. While the paper claims to introduce Frequency-based RoPE, this does not represent a significant novelty.

(2) Lack of respect for prior works:

   a. In reality, generating variable-length audio is not a difficult problem. The authors could have used AudioGen checkpoints, which can generate audio of any length. Additionally, this is not a critical issue; for instance, generating a 5-second dog bark versus a 10-second dog bark makes no practical difference. If the authors are focused on this, the first step should be to explain which types of sound actually require long audio. Why can't long audio simply be generated by stitching together sub-segments?

  b. The paper fails to compare with AudioGen [1]. Why? What are the advantages when compared to stable audio and AudioGen? Stable audio allows specifying start and end times, and AudioGen supports generating audio of any length.

  c. In the related work section, the authors write, "In TTA, a general framework is to use diffusion models pretrained on a large-scale audio dataset. A seminal work is AudioLDM..."  I question whether the authors have thoroughly reviewed AudioLDM's related work section. The authors really know the development of TTA? This lack of respect for prior works is unacceptable.

(3) Overclaims:
    The authors state, "FleXounDiT outperforms SOTA models with a significantly smaller model size." FleXounDiT is 612M, AudioLDM is 739M, and Make-an-Audio is 453M. How is 612M considered significantly smaller? Furthermore, the difference between 600M and 700M is negligible. Additionally, model size is not a significant issue in TTA tasks since the dataset size is manageable. Smaller models can be used without sacrificing performance.

(4) From the demo page, it is hard to judge the model is really good than previous works.

In summary, this paper presents numerous issues in terms of motivation, related work, methodology, and evaluation. It introduces little novelty. From my experience, this paper does not meet the standards of an ICLR conference submission.

[1] Kreuk F, Synnaeve G, Polyak A, et al. Audiogen: Textually guided audio generation[J]. ICLR 2023.

**Questions:**

Please refer to weakness part.

---

### Official Review · Reviewer_WfPA · 2024-10-29

**Soundness:** 3
**Presentation:** 2
**Contribution:** 2
**Rating:** 5
**Confidence:** 4

**Summary:**

FleXounDiT enables train-short-test-long (TSTL) ability of diffusion-based text-to-audio models by 1) using an absolute position embedding to frequency axis and RoPE to the temporal axis of the 2D latent feature of given mel spectrogram during training, and 2) using YaRN-based RoPE (Resonance RoPE) to a query-key scaling for length extrapolation at inference.

**Strengths:**

* The proposed method can potentially be applied to various TTA models based on Transformer-based layers with positional embeddings using 2D audio latent, where the one axis represents frequency and the other axis represents time.

* The method takes careful modification to the conventional RoPE-based method taking the frequency information into account. This is tailored to audio features unlike images.

**Weaknesses:**

* Since the proposed method is mostly about improving relative positional encodings for 2D latent-based TTA, readers may question if the existing models can benefit from the proposed method. Such ablation study is partly conducted in Table 3 and Figure 4(b), but the authors can consider measuring the effect of the method on top of baseline models as well. If the authors haven't found suitable baselines (i.e., 2D latent-based TTA with DiT), it can be mentioned more explicitly.

* The subjective scores (OVL and REL) do not contain confidence intervals. Without the details in how they measured subjective scores, the reviewer was not able to conclude if the difference is significant.

* Since their AudioCaps experiment did not use other training datasets, I think using AudioCaps focused models as baselines (AudioLDM2-AC-Large, for example) for the subjective evaluation would have been more appropriate.

**Questions:**

* Following the first item in Weakness section above, Stable Audio already has applied DiT to TTA with RoPE, but based on 1D VAE directly on waveform. I am not sure if the proposed method can be applied to such models.

* I am aware of the author's statements in the Appendix A: does Stable Audio's length extrapolation degrade when using off-shelf methods for RoPE, including the ones coverd in the paper?

* Line 348: TSCL -> TSTL?

---

### Official Review · Reviewer_9x2L · 2024-10-29

**Soundness:** 3
**Presentation:** 3
**Contribution:** 2
**Rating:** 5
**Confidence:** 4

**Summary:**

The paper introduces FleXounDiT, a framework that addresses the challenge of generating variable-length audio in text-to-audio (TTA) diffusion models. FleXounDiT utilizes an approach with relative position embeddings, allowing the model to generate audio of unseen lengths without additional tuning. This method also enables efficient training with shorter audio durations, thereby reducing computational costs while maintaining high performance. Empirical results demonstrate that FleXounDiT outperforms existing models in generating high-quality variable-length audio across benchmarks.

**Strengths:**

- The use of RoPE (Rotary Positional Embedding) is novel and effectively integrated, extending the model's ability to generate longer audio sequences. Using the standard RoPE resulted in decreased generation performance, but by proposing a Frequency-based RoPE tailored to the audio domain, they achieved improved generation performance.
- The frequency-based dynamic attention scaling technique shows promise in improving performance, especially for longer audio.
- The experimental results are adequate. The paper is well written. It combines clear, concise text with instructive visuals, making the methodology and results accessible and understandable.

**Weaknesses:**

- While U-Net-based architectures are traditionally suited for tasks requiring flexibility (due to CNN modules with fixed-weight kernels), it’s unclear why a diffusion transformer was chosen for this task. The advantages of using a diffusion transformer for variable-length audio generation compared to U-Net remain unclear.
- The evaluation relies solely on FAD for variable-length audio in Fig. 4 (a). To better assess robustness, additional metrics like KL divergence or CLAP should be included.
- Speech audio in the demo page, particularly male speech, lacks coherence and naturalness. The generated voice sounds inconsistent as if it is spoken by multiple individuals, which is a noticeable limitation compared to other TTA models.
- The addition of cross-attention module in the DiT Block needs justification and ablations. Is the MLP layer not sufficient?
- In 5.1 Implementation-dataets, "For evaluation on AudioCaps, we only focus on training and testing on the same dataset.". The model was trained only on AudioCaps to measure performance on this dataset. Could this be the reason why it shows lower FAD and higher CLAP scores in Table 1 compared to other models?

**Questions:**

There are several justifications / clarifications requested in the weaknesses sections. It would be beneficial for the authors to discuss them during the rebuttal.

**Details Of Ethics Concerns:**

No concern

---

### Official Review · Reviewer_bsti · 2024-11-03

**Soundness:** 3
**Presentation:** 3
**Contribution:** 3
**Rating:** 6
**Confidence:** 4

**Summary:**

The paper addresses the problem of extending audio length beyond what was covered during training for diffusion-based text-to-audio models, which is referred to as Train-Short-Test-Long (TSTL). The authors propose FleXounDiT, a diffusion model that is capable of generating sound events across variable durations. In the model, the authors introduce Rotary Position Embedding (RoPE) and an improved version of Resonance YaRN. The model is claimed to surpass SOTA models on 10-second benchmark, while having a smaller model size and being memory efficient during training.

**Strengths:**

The method is built up on the well-established framework of latent diffusion models, improving the performance on 10s benchmark and showing superior performance on shorter and longer samples. I think the capability to generate variable length audio is significant for a useful application and the literature has not been addressing this enough.

The main contribution of the paper is a RoPE that is absolute in the frequency axis and relative in the time axis, which is well motivated by the structure of mel-spectrogram and confirmed in the ablation study where replacing positional emb with RoPE does not work as well.

Overall, the paper is well written with solid experiments and ablation study. Authors compare the model performance with larger SOTA models and show significant improvements.

**Weaknesses:**

- Proposals in the paper aim to bring RoPE and context window extension from NLP to mel spectrogram generation as a single-channel 2D image. However, existing works show that it is possible to model mel spectrogram as a 1D signal and generate with a diffusion model [Make-an-Audio-2]. This is probably a better choice for variable length audio generation since we can directly applying techniques from NLP without frequency aware adaptation. Could authors explain why you choose to model this way?

- I find that the evaluation for variable length generation is inadequate with only FAD score reported. How do we know if FAD works well for 5s or 30s audio? I think subjective evaluation is needed to assess the coherence and continuity of the audio, which may be hard for FAD to reflect that on longer audio if the long audio is encoded into a single vector. I suggest that the authors may want to report mean FAD of a sliding 10s window with 1s step over the long audio.

**Questions:**

- In Fig 2 (left), the last indices are 2L-3, 2L- 2, 2L - 1 (is it typo?)
- How is $\mu$ in Eq 14. derived?

---

### Official Review · Reviewer_6Wfd · 2024-11-04

**Soundness:** 1
**Presentation:** 2
**Contribution:** 1
**Rating:** 3
**Confidence:** 5

**Summary:**

Implemented a text-to-audio DiT architecture and explored the impact of the positional encoding and its capability for variable-length generation.

**Strengths:**

Innovatively designed Freq-Rope2D for audio features, achieving zero-shot variable-length generation through techniques such as interpolation of positional encoding and attention scaling.

**Weaknesses:**

1: The comparison is unfair. In comparing different systems, Table 1 shows that Make-an-Audio2 exhibits the lowest FAD performance among all baselines, indicating its potential to achieve better audiocaps-test results when trained on Audiocaps-train; therefore, it should be compared after training on the same dataset.

Thus, the novelty seems limited: Note that achieving a model with better performance on audiocaps does not inherently require maintaining a fair dataset comparison. However, the main contribution of this paper should be the training-free extension of the 10-second model. Therefore, this data does not fully convince me of the necessity or novelty of your new model structure design. Aside from existing techniques, the technological innovation in this paper is the special position encoding (i.e. freq-rope) designed for audio signals, which heavily relies on your network architecture. It is very likely that under a simpler network architecture, such as methods that don't require frequency-domain patchify approaches (e.g., Make an audio 2, stable audio 2), these would clearly not require such a special freq-rope position encoding, as a 1-D position encoding would suffice. Therefore, controlling the consistency of the network architecture or ensuring fairness in comparison to the base version is crucial, as this is not a paper on DiT+audio generation. Thus, Table 1 needs to focus on fairness rather than superior performance.

2: Is the Rope mentioned in line 480 of the paper one-dimensional or two-dimensional? Even though the length in frequency does not change with audio duration, it is still possible to use a two-dimensional Rope for interpolation (only interpolating in the time dimension), but there is a lack of comparative experiments in this regard.

3: Table 2 should be the most important experiment of this paper, but since their base models have different performances at 10 seconds, comparing their FAD values at longer durations is meaningless. Additionally, the metrics are too singular. Referring to the testing scheme of syncdiffusion[1], the focus should be on evaluating the generated quality from multiple perspectives.

4: Section 5.4 lacks comparisons with other models' performances.

5: In Figure 5, the middle section of the 20-second sample seems to present an informationless state.

[1]:https://arxiv.org/abs/2306.05178

**Questions:**

1:There was no attempt to evaluate the performance of solutions beyond 30 seconds, and now directly training a 30-second model is no longer constrained by GPU memory. Is there still significance in extending to 10 seconds?

2:From listening to the demo, the generation of vocals seems to tend toward chaos, and the sound quality appears to be relatively low compared to models like make-an-audio2 and Stable Audio Open.
From my perspective, if the 10-second audio is extended to 30 seconds, the three sound events should blend organically, rather than generating the first two within the 10 seconds and then extending the third for an additional 20 seconds (I observed such examples in the demo). I understand that due to differences in training data, the sound quality may be difficult to control, but in audio generation, the most important aspect is temporal balance, not just the order of events. It seems that this model may not have handled this very well.

---

### Note · Authors · 2024-11-15

I have read and agree with the venue's withdrawal policy on behalf of myself and my co-authors.